# Hepatitis B Virus DNA is a Substrate for the cGAS/STING Pathway but is not Sensed in Infected Hepatocytes

**DOI:** 10.3390/v12060592

**Published:** 2020-05-29

**Authors:** Lise Lauterbach-Rivière, Maïwenn Bergez, Saskia Mönch, Bingqian Qu, Maximilian Riess, Florian W. R. Vondran, Juliane Liese, Veit Hornung, Stephan Urban, Renate König

**Affiliations:** 1Host–Pathogen Interactions, Paul-Ehrlich-Institut, 63225 Langen, Germany; maiwenn.bergez@icloud.com (M.B.); saskia.moench@pei.de (S.M.); max.riess@yahoo.de (M.R.); 2Department of Infectious Diseases, Molecular Virology, University Hospital Heidelberg, 69120 Heidelberg, Germany; qubingqian@gmail.com (B.Q.); stephan.urban@med.uni-heidelberg.de (S.U.); 3Regenerative Medicine and Experimental Surgery (ReMediES), Department of General, Visceral and Transplant Surgery, Hannover Medical School, 30625 Hannover, Germany; Vondran.Florian@mh-hannover.de; 4German Centre for Infection Research (DZIF), Hannover-Braunschweig, 38124 Braunschweig, Germany; 5Department of General and Thoracic Surgery, Justus-Liebig-University Giessen, 35392 Giessen, Germany; Juliane.Liese@chiru.med.uni-giessen.de; 6Gene Center and Department of Biochemistry, Ludwig-Maximilians-Universität München, 81377 Munich, Germany; hornung@genzentrum.lmu.de; 7German Center for Infection Research (DZIF), 69120 Heidelberg, Germany; 8German Center for Infection Research (DZIF), 63225 Langen, Germany; 9Sanford Burnham Prebys Medical Discovery Institute, Immunity and Pathogenesis Program, La Jolla, CA 92037, USA

**Keywords:** hepatitis B virus (HBV), innate immunity, STING, cGAS, innate immune sensing, interferon response

## Abstract

Hepatitis B virus (HBV) chronic infection is a critical risk factor for hepatocellular carcinoma. The innate immune response to HBV infection is a matter of debate. In particular, viral escape mechanisms are poorly understood. Our study reveals that HBV RNAs are not immunostimulatory in immunocompetent myeloid cells. In contrast, HBV DNA from viral particles and DNA replication intermediates are immunostimulatory and sensed by cyclic GMP-AMP Synthase (cGAS) and Stimulator of Interferon Genes (STING). We show that primary human hepatocytes express DNA sensors to reduced levels compared to myeloid cells. Nevertheless, hepatocytes can respond to HBV relaxed-circular DNA (rcDNA), when transfected in sufficient amounts, but not to HBV infection. Finally, our data suggest that HBV infection does not actively inhibit the DNA-sensing pathway. In conclusion, in infected hepatocytes, HBV passively evades recognition by cellular sensors of nucleic acids by (i) producing non-immunostimulatory RNAs, (ii) avoiding sensing of its DNAs by cGAS/STING without active inhibition of the pathway.

## 1. Introduction

Innate immunity is the first line of defense against pathogens. Detection of pathogen-associated molecular patterns (PAMPs) by cellular pathogen recognition receptors (PRRs) triggers signaling pathways leading to interferon (IFN) production, which induces antiviral interferon-stimulated genes (ISGs) and pro-inflammatory cytokines.

In viral infections, viral RNA or DNA are common PAMPs that can be detected by cytosolic PRRs. Members of the RIG-I-like Receptor (RLR) family such as Retinoic Acid Inducible Gene I (RIG-I) and Melanoma Differentiation-Associated protein 5 (MDA5) sense foreign RNAs and activate Mitochondrial Antiviral Signaling Protein (MAVS), while foreign viral DNA is recognized by sensors like cyclic GMP-AMP Synthase (cGAS). Activated cGAS produces 2’3’-cyclic GMP-AMP (cGAMP), which activates Stimulator of Interferon Genes (STING). STING or MAVS activation can both lead to the activation of Interferon Regulatory Factor 3 (IRF3), which promotes IFN gene transcription [1]. To evade antiviral innate responses, viruses have evolved escape strategies involving the inhibition of innate signaling pathways by viral proteins, or the shielding of the viral genome from innate sensors (reviewed in [1,2]).

To date, more than 250 million people are chronically infected with hepatitis B virus (HBV), which is a high risk factor for liver cirrhosis and hepatocellular carcinoma. Its interplay with the signaling pathways leading to IFN production is still a matter of debate. Some publications have described a type I and III IFN or pro-inflammatory response to HBV infection in cultured hepatocytes [3,4,5,6,7]. HBV may also induce pro-inflammatory cytokines in immune cells such as Kupffer cells, the liver macrophages, even if they are not productively infected [8,9,10]. The induction of an innate response to HBV in infected hepatocytes has however been questioned by other reports [9,11,12,13,14], in line with initial studies in chimpanzees [15] and patients [16,17]. Therefore, HBV has been proposed to be a stealth virus in infected hepatocytes, but the mechanisms behind the lack of innate immune responses are not fully understood. 

HBV particles contain a partially double-stranded DNA, the relaxed-circular DNA (rcDNA), which is repaired into a covalently closed circular DNA (cccDNA) in the nuclei of infected cells. The cccDNA is then transcribed into mRNAs and pregenomic RNA (pgRNA) that are exported to the cytoplasm. The pgRNA is encapsidated into newly synthesized capsids, where it is reverse transcribed into rcDNA. Both HBV RNAs and DNAs are therefore potential PAMPs. HBV pgRNA has been proposed to be sensed by the RIG-I- [4] or MDA5- pathways [3]. Surprisingly, MDA5 was also reported to bind HBV DNA [3]. Recently, sensing of HBV DNA by the cGAS/STING pathway was suggested [11,18] but the expression and functionality of this pathway in hepatocytes is unclear [9,11,14,18,19]. Therefore, the immunostimulatory potential of HBV nucleic acids and the PRR involved require further investigation.

HBV could inhibit the innate immune response to escape its antiviral effects, as suggested by several publications [6,11,20,21,22,23,24,25,26,27,28,29]. In particular, the regulatory Hepatitis B virus X protein (HBx) has been described to inhibit the MAVS pathway [21,27], while the viral polymerase was reported to inhibit STING [30] and the phosphorylation of IRF3 [24,25]. Verrier et al. [11] suggested the down-regulation of cGAS, STING and TANK-binding kinase 1 (TBK1) by HBV [11]. On the other hand, several publications claimed an absence of inhibition of the innate response by HBV [9,12,31,32]. Most of them tested several RNA-sensing pathways but the DNA-sensing pathway has been less extensively investigated. 

In this study, we first assessed the immunostimulatory potential of naked HBV DNAs and RNAs in monocyte-derived dendritic cells (MDDCs) as a model for highly immunocompetent cells. For the first time, we could demonstrate that HBV RNAs are not immunostimulatory. On the contrary, naked HBV DNA can elicit a strong innate immune response, mediated by the cGAS/STING pathway. In hepatocytes, this pathway is expressed at a low level but retains its ability to sense naked HBV DNA, when present in sufficient amounts. However, this pathway is not activated upon productive HBV infection although our data suggest that the virus does not actively suppress it.

## 2. Materials and Methods 

### 2.1. Cells and Ethics Statements

HepG2-hNTCP cells and HepaRG-hNTCP cells are derived from HepG2 cells and from HepaRG cells respectively and have been stably transduced with a lentiviral vector expressing HBV receptor, the sodium-taurocholate cotransporting polypeptide (NTCP) [33]. HepG2.2.15 cells are derived from HepG2 cells and have been stably transfected with HBV genome [34]. HepG2 H1.3deltaX cells are derived from HepG2 cells and contain the stable integration of a 1.3-fold HBV genome carrying premature stop codon mutations in both 5’ and 3’ parts of the HBx open reading frame [35]. HepAD38 cells are derived from HepG2 cells and contain a stable integration of an HBV genome under the control of a tetracycline repressor promoter [36]. In the absence of tetracycline these cells express all HBV RNAs and produce HBV particles. HepG2-hNTCP, HepAD38 and HepG2 H1.3deltaX were maintained in Dulbecco’s modified eagle medium (DMEM) containing 10% fetal calf serum (FCS) and 2 mM L-Glutamine (complete DMEM). HepaRG-hNTCP were grown in William’s E medium supplemented with 10% FCS, 2 mM L-glutamine, 100 U/mL penicillin, 100 µg/mL streptomycin, 5 × 10^−5^ M hydrocortisone hemisuccinate and 5 μg/mL insulin. 

THP-1 cells are a human monocytic leukaemia cell line and can be differentiated into macrophage-like cells using phorbol 12-myristate 13-acetate (PMA) [37]. THP-1 cells deficient for cGAS (CGAS), STING (TMEM173) and MAVS (MAVS) were previously described [38]. THP-1 cells were cultivated in RPMI supplemented with 10% fetal calf serum (FCS) and 2 mM L-Glutamine. For differentiation, 40 ng/mL of PMA were added for 48 h. 

Cell lines were authenticated as described in the Table 1 and regularly tested to exclude any mycoplasma contamination using a PCR-based method with the PCR Mycoplasma Test Kit II from AppliChem.

Primary human hepatocytes (PHH) were isolated from liver specimens resected from patients undergoing partial hepatectomy because of hepatocellular carcinoma or severe cirrhosis. The patients were tested negative for HBV, Hepatitis C virus (HCV) and Human immunodeficiency virus (HIV). Approval from the local and national ethics committees (Ethic committee from the Goethe University of Frankfurt, agreement number 343/13, and the Hannover medical school, agreement number 252-2008) and informed consent from patients were obtained. PHH were isolated with a two-step perfusion method and cultured as described previously [40]. Alternatively, cryopreserved plateable PHH were purchased from Thermo Fisher Scientific (Waltham, Massachusetts, USA). These cryopreserved PHH were isolated from dead patients who were tested negative for HBV, HCV, HIV-1 and 2, Human T- cell leukemia virus (HTLV) 1 and 2 and Cytomegalovirus (CMV). PHH were maintained in Williams’ Medium E supplemented with Serum-free Hepatocyte Maintenance Supplement Pack (Thermo Fisher Scientific CM4000) according to the provider’s instructions. 

Cryopreserved human Kupffer cells were purchased from Thermo Fisher Scientific (Waltham, Massachusetts, USA) and handled according to the provider’s instructions.

Monocyte-derived dendritic cells (MDDCs) or monocyte-derived macrophages (MDMs) were generated from human peripheral blood mononuclear cells (PBMC) using respectively 25 U/mL GM-CSF (Leukine^®^Sargramostim, Genzyme, Cambridge, MA, USA) and 800 U/mL IL-4 (PeproTech, Rocky Hill, NJ, USA) or 50 U/mL GM-SCF. The PBMCs were isolated from human buffy coats of anonymous blood donors purchased from the German Red Cross Blood Donor Service Baden-Württemberg Hessen, Germany.

### 2.2. HepG2-hNTCP Overexpressing cGAS and STING

HepG2-hNTCP were transduced with pLX304-based lentiviral vectors encoding V5-tagged cGAS and selected with 20 µg/mL of blasticidin. For each experiment, the selected cells were freshly transduced with pLX304-based lentiviral vectors encoding V5-tagged STING and were used 6 days post-transduction for experiments.

### 2.3. Hepatitis B Virus Production and Infection

Preparation of Hepatitis B Virus wild type (HBVwt) inoculum from HepAD38 cells or HBV deficient for HBx (HBV X-) from HepG2 H1.3deltaX was performed as previously described [41]. Viral particles were titered by qPCR quantification of viral RC-DNA with the primers HBV_RC_F and HBV_RC_R (Table 2). For infection, HepG2-hNTCP and PHH were seeded on collagen-coated plates. HBV infections were performed in presence of 4% Polyethylene glycol 8000 and 2% DMSO unless otherwise indicated. When indicated, HBV entry inhibitor Myrcludex B [42] (1 µM) was added. Twenty-four hours after viral inoculation, the medium was changed and complete DMEM supplemented with 2% DMSO was added.

### 2.4. Other Viruses and Viral Vectors

Sendai virus, strain Cantell, was provided by G. Kochs, Medical Center-University of Freiburg, Germany. 

The DNA virus MVA-gfp based on Modified vaccinia Ankara was kindly provided by Gerd Sutter, GSF-Institute for Molecular Virology, Munich, Germany [43,44]. 

### 2.5. Quantitative Reverse Transcription PCR (RT-qPCR)

For RT-qPCR analysis, total RNAs were extracted from cell pellets using NucleoSpin^®^ RNA Plus Kit (Macherey Nagel, Düren, Germany). RT-qPCRs were performed on a LightCycler 480 (Roche, Basel, Switzerland) using QuantiTect SYBR Green RT-PCR Kit (Qiagen, Hilden, Germany) and specific primers (Table 2). For analysis of HBV RNAs, primer sets that specifically amplify the 3.5 kb, 2.4 kb and 2.1 kb HBV RNAs (HBV RNA 3 F, HBV RNA 3 R, Table 2) were used. Unless otherwise stated, Ribosomal Protein L13 A (RPL13A) was used as a reference gene to normalize the samples. For each sample, the relative amount of RNA was calculated using the formula 2-ΔCt with ΔCt = (Ct gene of interest − Ct RPL13A). When indicated, relative change to the control condition was calculated using the formula 2^- ΔΔ Ct^ with ΔΔCt = ΔCt sample − ΔCt control.

For Figure 3, the expression levels of the genes of interest and of 2 reference genes, RPL13A and TBP (TATA-box binding protein), were calculated using a standard curve generated with serial dilutions of THP1 RNA. For each sample, the values of the genes of interest were normalized to the geometric mean of the 2 reference genes. The limit of detection for each sample and each gene of interest was calculated as the RNA amount measured in the last detectable standard dilution of THP1 for the gene of interest divided by the geometric mean of the expression levels of the 2 reference genes for each sample.

### 2.6. Reagents

2′3′ cGAMP was purchased from Invivogen (San Diego, CA, USA) and used at a final concentration of 2 µg/mL in cell supernatant. HT-DNA (herring testis DNA) was purchased from Sigma-Aldrich (St. Louis, MO, USA).

### 2.7. Preparation and Quantification of Viral Nucleic Acids for Transfection Experiments

For viral nucleic acid transfection experiments, total nucleic acids from HBV, Sendai Virus (SeV) and Modified vaccinia Ankara (MVA)-gfp viral stocks were extracted using the High Pure Viral Nucleic Acid Kit (Roche, Basel, Switzerland). When indicated, samples were digested with 1 mg/mL of RNAseA for 1 h at 37 °C followed by 15 min inactivation at 72 °C and an addition of RNAseIN (Promega, Madison, WI, USA), or with DNAse using the Turbo DNA-free kit (Ambion, Thermo Fisher Scientific, Waltham, MA, USA) accordingly to the manufacturer’s instructions. 

Total RNAs from HepAD38 or HepG2 cells were extracted using the NucleoSpin^®^ RNA Plus Kit (Macherey Nagel, Düren, Germany) and any DNA contaminants were digested with the Turbo DNA-free kit (Ambion, Thermo Fisher Scientific, Waltham, Massachusetts, USA).

HBV replication intermediates were extracted from the cytoplasmic fraction of HepAD38 cells as described in [45] with minor modifications. Cells were incubated for 45 min on ice in hypotonic buffer (10 mM Hepes pH 8, 1.5 mM MgCl2, 10 mM KCl). One percent of NP-40 was then added and the samples were vortexed for 30 s. Nuclei were pelleted by centrifugation at 6000 g at 4 °C for 5 min. The supernatants (cytoplasmic fractions) were treated with 8U/ml of DNAseI in presence of 3.3 mM of MgOAc for 90 min at 37 °C in order to keep only encapsidated DNA. After proteinase K digestion, the DNA was extracted with phenol-chloroform and treated with RNAseA as described earlier.

HBV DNA from particles, replication intermediates and MVA-gfp DNA copy numbers were determined by qPCR, HBV RNA and SeV RNA copy numbers were quantified using one-step reverse transcription qPCR (RT-qPCR) using specific primers and serial dilutions of standard DNA oligonucleotides (Table 2). For RNA samples, the results are expressed as cDNA-equivalent since the copy number of the viral RNAs is assumed to be proportional to the respective specific cDNA quantification. RT-qPCRs and qPCRs were performed using QuantiTect SYBR Green RT-PCR Kit (Qiagen, Hilden, Germany) respectively with and without RT on a LightCycler 480 (Roche, Basel, Switzerland).

### 2.8. Transfections

Viral nucleic acids or HT-DNA (Sigma-Aldrich, St. Louis, MO, USA) were transfected into MDDCs or PMA-differentiated THP1 using Lipofectamine 2000 (Thermo Fisher Scientific, Waltham, MA, USA) or in PHH and HepG2-hNTCP-derived cell lines using X-tremeGENE HP reagent (Merck, Darmstadt, Germany) according to the manufacturer’s instructions.

For MDDC transfection experiments (Figure 1), Table 3 indicates the copy number/cell of viral DNAs (DNA copy/cell, determined by qPCR) or of viral RNAs (expressed as cDNA-equivalent copy/cell, determined by RT-qPCR). The indicated copy numbers/cell refers to the undiluted nucleic acids. SeV RNAs, MVA-gfp DNA and HBV DNA from viral particles were quantified from undigested nucleic acid preparations. HBV DNA replication intermediates were quantified from RNAse-digested samples. HBV RNA from viral particles or from total HepAD38 RNAs were quantified from DNase-digested samples to avoid any DNA contamination, and a control without RT was performed.

### 2.9. Western Blots

Cell samples were lysed using a TritonX-100 based Lysis Buffer (100 mM NaCl, 10 mM EDTA, 20 mM TRIS pH7.5, 1% TritonX-100, 1% Sodiumdesoxycholate). Proteins were separated by SDS-PAGE and transferred to a nitrocellulose membrane. Specific proteins were detected using the antibodies listed in the Table 4. Secondary antibodies conjugated to the horseradish peroxidase (HRP; GE Healthcare, Chicago, IL, USA) were used for chemiluminescence detection.

### 2.10. IRF3 Nuclear Translocation Assays

HepG2-hNTCP overexpressing cGAS and STING were infected or not with HBV (MOI 10000) in the presence of 2% DMSO. Four days post infection (dpi), the DMSO was removed. Six dpi, the cells were seeded on collagen-coated coverslips in 48-well plates and one day later transfected or not with 2 µg/well of HT-DNA. Sixteen hours post-transfection the samples were stained for IRF3 and HBV core (HBc) and imaged with a Zeiss LSM 510-META Confocal laser scanning microscope (Oberkochen) as described below.

### 2.11. Immunofluorescence

Cells were fixed with 4% paraformaldehyde for 15 min, washed 3 times with PBS, permeabilized with 0.25% Triton-X100 for 7 min at room temperature and blocked for 1 h in PBS-0.1% Tween (PBST) containing 5% of bovine serum albumin (BSA). The coverslips were then incubated for 2 h at room temperature with the primary antibodies, washed 3 times in PBST, and incubated for 1 hour with the secondary antibodies (Table 5). The coverslips were washed 3 times in PBST and the DNA was stained with Hoechst reagent. 

### 2.12. Microscope Image Acquisition

For the IRF3 translocation assay (Figure 6) the samples were mounted with Mowiol medium. They were imaged using a Zeiss LSM 510-META Axiovert 200M Confocal laser scanning microscope, with a 40× objective (EC "Plan-Neofluar" 40×/1,30 Oil DIC) and an Axiocam camera with the acquisition software ZEN 2009, Version: 5.5.0.451.

Imaging of the MVA-gfp-infected HepG2-hNTCP (Appendix A) was performed in the culture plates with cell medium using a Nikon Eclipse TS100 inverted microscope with a 10× objective (Achromat, 10×/0,25 Ph1 DL) and a Nikon DS-Vi1 camera, with the acquisition software NIS-Elements F 4.30.00 (Build 1020) 64 bit.

For HBcAg staining in PHH (Appendix A), stained cells were imaged in PBS without mounting medium using an LEICA DM IRB microscope with a 20× objective (N PLAN L 20*/0.40 CORR PH 1) and am HAMAMATSU camera with the acquisition software HoKawo 2.12.

All microscope acquisitions were performed at room temperature. Contrast/brightness of the images were adjusted with Fiji ImageJ (version 1.52p; open source software).

## 3. Results

### 3.1. HBV DNAs but not RNAs are Immunostimulatory

We first investigated whether different species of HBV nucleic acids (DNAs or RNAs from viral particles or from HBV-producing cells) are immunostimulatory. To this aim, we used monocyte-derived dendritic cells (MDDCs) as a model for highly sensitive immune cells that most likely express all nucleic acid sensors. The MDDCs were transfected with ten-fold serial dilutions of quantified viral nucleic acids and ISG54 mRNA induction was measured by RT-qPCR. ISG54 mRNA was chosen as a marker for activation of nucleic acid-sensing pathways, as it is a direct target gene of IRF3-dependent transcription [46,47]. The copy number of the viral nucleic acids (DNA or RNA) used for MDDC transfection was determined as described in the material and methods section and is indicated in Table 3. 

To validate the system, we first used total nucleic acids isolated from viral preparations of the RNA virus SeV or of a Gfp-expressing vector based on the DNA virus Modified vaccinia Ankara (MVA-gfp). SeV RNA activates RIG-I signaling and is used as a model to stimulate cytoplasmic sensing of RNA [48]. MVA-gfp DNA activates cGAS/STING signaling and is employed as a model to stimulate cytoplasmic sensing of DNA [49]. We observed a high ISG54 induction upon transfection of MDDCs with SeV nucleic acids (Figure 1) demonstrating the functionality of the RIG-I pathway. This response is abrogated upon RNAse digestion of SeV nucleic acids, while it is unaffected by DNAse digestion, proving that this response is specific for RNA. Nucleic acids from MVA-gfp particles also induce a dose-dependent ISG54 response, validating the functionality of the cGAS/STING pathway.

Interestingly, nucleic acids extracted from HBV viral preparations induced an ISG54 response, which was unaffected by RNAse digestion but is abrogated by DNAse digestion (Figure 1), demonstrating that HBV DNA contained in the viral particles was immunostimulatory. HBV viral preparations also contained small amounts of HBV RNAs (Table 3), confirming the reports by Cheng et al. [9] However, DNAse-digested nucleic acids from HBV particles do not induce an ISG54 response (Figure 1), suggesting that these particle-associated HBV RNAs do not account for the immunostimulatory activity of HBV nucleic acids from particles.

In HBV-infected cells, various DNA replication intermediates are produced in the cytoplasm upon reverse-transcription of the pgRNA [50] and may also act as PAMPs. To test their immunostimulatory potential, we extracted HBV DNA replication intermediates from the cytoplasmic fraction of HepAD38 cells after removal of any cellular or viral RNA contamination using RNAse (as described in Materials and Methods and in [45]). HepAD38 contain an integrated copy of the HBV genome and produce all viral RNAs and DNA replication intermediates, as well as functional HBV particles [36]. When HBV DNA replication intermediates were transfected into MDDCs, the kinetic of ISG54 induction varied depending on the donors. The analysis at 6 h (Figure 1) shows a slight but not statistically significant ISG54 induction. However, when a second time point was included in the analysis (6 h + 24 h, Appendix A), a significant ISG54 induction was observed with the highest concentration (5000 copies/cell) of HBV replication intermediates. Intriguingly, the response was apparently weaker than when using the same copy number (5000 copies/cell) of HBV rcDNA (Figure 1). However, we cannot exclude that different DNA structures are transfected with different efficiencies, leading to apparent differences in their immunostimulatory potential. We therefore conclude that HBV replication intermediates are immunostimulatory but we cannot affirm that the difference with HBV DNA from viral particles is biologically relevant.

Furthermore, we tested the immunostimulatory potential of HBV RNAs from HBV-producing cells, using total RNAs from HepAD38 cells. However, no ISG54 induction was detected upon transfection of 1.3 × 10^4^ cDNA-equivalent copies/cells of HepAD38 RNAs into MDDCs at 6 h or 24 h (Figure 1, Appendix A and Table 3). Considering that only 28 cDNA-equivalent copies/cells of SeV RNAs induce a robust ISG54 response at 6 h in MDDCs (Figure 1, undiluted sample, Table 3), which is 4.6 × 10^2^ times less than the amount of HBV RNAs used, we conclude that that HBV RNAs (mRNAs or pgRNA) are not immunostimulatory. 

In summary, naked HBV DNA from particles and DNA replication intermediates from HBV-producing cells have the potential to elicit an innate response whereas HBV RNAs from HBV-producing cells are not immunostimulatory.

### 3.2. Virion-Associated HBV DNA is Sensed by the cGAS/STING Pathway

To identify which PRRs and pathways sense and respond to HBV DNA, we used a panel of THP-1 knock-out (KO) cell lines deficient for key nodes of the sensing pathways, cGAS, STING or MAVS (THP-1 wt, ΔSTING, ΔcGAS and ΔMAVS respectively, Figure 2A). As expected, KO of STING or cGAS did not significantly affect the ISG54 response to SeV infection, while MAVS KO abrogated it (Figure 2B). On the contrary, STING or cGAS KOs abrogated the innate response to transfection with the cGAS agonist herring testes DNA (HT-DNA) while the transfected DNA was well sensed by MAVS KO, proving the validity of the chosen assay system. Similar to MDDCs, transfection of HBV nucleic acids from viral particles in THP1 wt strongly induced a dose-dependent ISG54 response, which was abrogated by DNAse digestion. Interestingly, STING and cGAS KOs totally abrogated the response to HBV nucleic acids, while MAVS KO had no significant effect. These results indicate that HBV DNA is sensed through the cGAS/STING pathway, while the RLR pathway is not involved. In addition, this further confirms that virion-associated HBV RNAs do not account for the innate response.

### 3.3. Hepatocytes Express Low Levels of the DNA Sensors Compared to Immune Cells

Having proven that HBV DNA is able to stimulate the cGAS/STING pathway, we analyzed the expression of this pathway in hepatocytes. To this aim, we first measured by RT-qPCR the RNA level of cGAS, STING as well as other components of PRR pathways in hepatic cell lines (HepG2 and HepaRG, overexpressing or not the viral receptor NTCP) and in PHH and compared them to primary immune cells (MDDCs, monocyte-derived macrophages (MDM), Kupffer cells) and THP1 cells (Figure 3). cGAS RNAs are expressed to a similar level in all the tested immune cells, however its expression is strongly reduced in hepatic cells although still detectable in PHH. STING shows a variable expression pattern in immune cells (high in THP1, MDMs and Kupffer cells, lower in MDDCs as seen earlier in [51]). In hepatic cells, STING expression is generally lower than in MDDCs, although it is always above the detection limit. Gamma-interferon-inducible protein 16 (IFI16), proposed as an immune sensor of retroviral DNA intermediates and a cGAS cooperative factor for activating STING [52], is expressed in PHH and HepaRG-hNTCP, although it is slightly reduced compared to immune cells, but is undetectable in the HepG2-derived cell lines. PQBP1, a cofactor of the cGAS/STING pathway involved in sensing HIV-1 reverse-transcribed DNA [53], is expressed to similar levels in all tested cell types. Finally, RIG-I and MAVS are expressed to similar levels in immune and hepatic cells. In conclusion, some components of the DNA-sensing pathways, and in particular the cGAS/STING pathway are considerably less expressed in hepatocytes than in immune cells, while components of the RNA-sensing pathways are expressed to comparable levels.

### 3.4. Hepatocytes are Competent for Sensing HBV DNA

We next tested whether the low levels of cGAS/STING correlate with an altered functionality of the pathway in hepatocytes. To this aim, we first stimulated HepG2-hNTCP with the STING agonist cGAMP (Figure 4A). Interestingly, ISG54 was transiently but significantly induced in HepG2-hNTCP eight hours after cGAMP stimulation, indicating the functionality of STING and of the downstream pathway. We next tested the complete cGAS/STING pathway using infection with MVA-gfp, or transfection with HT-DNA and with HBV nucleic acids from viral particles. To determine if the levels of cGAS and STING are limiting for DNA sensing, we overexpressed cGAS and STING in HepG2-hNTCP using lentiviral vectors and compared them with cells transduced with a control vector (Figure 4B). The infection efficiency of MVA-gfp was controlled by the GFP expression (Appendix A). Since the transfection reagent alone induced a weak ISG54 response compared to an untransfected control, we compared the transfected samples to a mock-transfected control (transfection reagent only), while the samples infected with SeV or MVA, which have not been treated with a transfection reagent, were compared to an uninfected/untransfected control (mock). Interestingly, the control-transduced HepG2-hNTCP cells were able to significantly respond to transfection with HBV nucleic acids from viral particles (Figure 4B). The response to HBV nucleic acids transfection was dose-dependent (1500 copies/cell induce ISG54 by 2 log, while 150 copies/cells are insufficient), suggesting that the amount of foreign DNA in hepatocytes needs to reach a threshold to activate the DNA-sensing pathway. This induction was abrogated by DNAse digestion, demonstrating the specificity of the response for DNA. Transduction with cGAS and STING vectors increased the RNA level of cGAS and STING by 1 to 2 log compared to the control vector (Appendix A) and enhanced the response to all DNA stimuli but not to SeV infection. This shows that HepG2-hNTCP are able to sense HBV DNA from particles, to some extent, but cGAS and STING expressions are limiting.

We next assessed the functionality of the DNA-sensing pathway in PHH (Figure 4C). Interestingly, transfection of PHH with undigested, but not with DNAse-digested HBV DNA (1500 copies/cell), induced ISG54 in all four donors, indicating that PHH are able to sense HBV DNA and that the response is specific for DNA.

### 3.5. HBV DNA is not Sensed During Productive Infection of Hepatocytes

In parallel, we analyzed the activation of the nucleic acid-sensing pathways upon productive infection of hepatocytes with HBV (Figure 5). In addition to ISG54 induction, we also investigated IRF3 activation by phosphorylation, as an upstream event common to both RNA and DNA cytosolic-sensing pathways. IFN-λ1 mRNA expression, previously proposed to be induced by HBV [4,5], was also analyzed. Although HepG2-hNTCP and PHH were efficiently infected, as demonstrated by the accumulation of HBV RNAs over time, we observed no induction of ISG54 or IFN-λ1 (5A and 5B). No IRF3 phosphorylation was detected upon HBVwt infection of PHH (Figure 5C). In addition, depletion of the viral HBx protein (HBV X- mutant), previously proposed to inhibit IRF3-dependant innate-sensing pathways [21,27], did not restore the activation of the pathway in infected PHH (Figure 5B,C). However, the HBV X- mutant replicates less efficiently than HBVwt. Therefore we cannot rule out that higher MOIs of HBV X- may induce an innate response. In contrast, infection with SeV induces a robust innate response in these cells. 

In conclusion, hepatocytes are able to sense and respond to naked HBV DNA but we confirmed that this viral DNA is not sensed upon productive HBV infection. Furthermore, the absence of IRF3 phosphorylation in HBV-infected PHH suggests either an absence of viral DNA sensing by PRRs or, alternatively, an inhibition of the signaling pathways upstream of IRF3 phosphorylation. Alternatively, it is possible that the amount of HBV DNA present in infected hepatocytes is not sufficient to be sensed, since the efficiency of the cGAS/STING pathway is poor in these cells.

### 3.6. HBV Does not Inhibit the Innate Immune Response to Foreign DNA in Hepatocytes

The hypothesis of an inhibition of the DNA-sensing pathway by the virus is still a matter of debate. Previous studies showed an inhibition of STING and IRF3 phosphorylation by HBV polymerase [24,30] and the down-regulation of cGAS and STING upon HBV infection [11]. On the contrary, another report concluded on the absence of an inhibition of the DNA-sensing pathway upon HBV expression in HepAD38 cells [12]. We therefore addressed this controversial question in HBV-infected HepG2-hNTCP. In order to focus only on infected cells, we immunostained HBc as an infection marker, and analyzed IRF3 nuclear translocation after HT-DNA transfection. IRF3 is translocated to the nucleus upon stimulation of the cGAS/STING pathway. To get a higher percentage of IRF3 nuclear translocation, we used HepG2-hNTCP overexpressing cGAS and STING (Appendix A). No nuclear staining was observed for IRF3 in the unstimulated cells, independent of HBV infection (Figure 6; row “HBV untransfected”), consistent with the lack of an innate response to HBV infection. In the mock samples, HT-DNA transfection induced IRF3 nuclear translocation in 16.4% of the cells (Figure 6B, graph). We observed a similar rate of IRF3 translocation in HBc-positive, HBV-infected cells transfected with HT-DNA (19.4%). The ratio +/- standard deviation of IRF3 nuclear translocation in (HT-DNA-transfected HBV-infected cells) / (HT-DNA-transfected mock-infected cells) is 1.18 +/- 0.55. This indicates that HBV infection does not inhibit the innate immune response to foreign DNA in HepG2-hNTCP overexpressing cGAS and STING. However, because of the low percentage of IRF3 nuclear localization upon HT-DNA transfection in the absence of cGAS/STING overexpression, we could not perform this experiment in PHH. Therefore, we cannot confirm that HBV does not interfere with the DNA-sensing pathway in cells expressing a low level of sensors.

In addition, we measured the level of cGAS and STING mRNAs in mock-infected versus HBV-infected PHH (MOI 5000). Immunofluorescence staining of HBc indicated that the majority of the cells were infected (Appendix A). However, productive HBV infection did not affect the levels of cGAS or STING compared to uninfected PHH or PHH infected in the presence of the entry inhibitor Myrcludex B (Appendix A).

## 4. Discussion

In this study, we investigated the evasion of HBV from nucleic acid sensors during infection of hepatocytes. We first demonstrated that naked HBV RNAs are not immunostimulatory. In contrast, we found that naked HBV DNAs, and particularly the DNA from viral particles, can be sensed by the cGAS/STING pathway. We showed that in hepatocytes, including PHH, this pathway is expressed at a low level, but is able to respond to high amounts of transfected HBV DNA associated with particles, although hepatocytes do not respond to productive HBV infection. We further demonstrated that HBV does not actively inhibit the innate response to foreign DNA in infected hepatocytes overexpressing cGAS and STING. 

We showed for the first time that naked HBV RNAs, including mRNAs and the pgRNA, are not immunostimulatory in transfected MDDCs, used as a model for highly immune competent cells. All HBV RNAs are transcribed by the cellular RNA polymerase Pol II, similar to cellular mRNAs. It is known that cellular mRNAs bear specific features that avoid their recognition by the RLR sensors [54,55]. Binding to the RLR sensor, RIG-I is inhibited by a capping of the cellular mRNA, both by the 7-methyl guanosine cap (Cap0) and 2’-O-methylation (Cap1) [56]. Interestingly, HBV RNAs have been described to bear a Cap0 [57,58]. The presence of a Cap1 on HBV RNAs has not been formally studied, but could account additionally for the absence of recognition by RIG-I. In addition, HBV RNAs contain N6-methyladenosine (m6A) RNA methylation [59], which has been shown to impair RIG-I binding [60]. In addition, we demonstrated that HBV RNA species are associated with HBV particles, confirming observation from Cheng et al. [9], but that they do not account for the observed ISG54 induction when MDDCs are transfected with HBV nucleic acids from particles (Figure 1). The origin of these particle-associated RNAs is unclear and may arise from incomplete reverse-transcription of the pgRNA. Alternatively, RNAs from HBV-producing cells may remain associated with the viral particles during the preparation of the viral stocks. 

On the contrary, we demonstrated that HBV DNA isolated from viral particles is immunostimulatory in myeloid as well as hepatic cells including PHH. The majority of the HBV DNA represents the most likely rcDNA as it is known that mature nucleocapsids are secreted as double-stranded DNA-containing virions [61]. However, we cannot exclude that also immature virions containing single-stranded DNA have been released as it was observed with certain HBc mutants [62]. CRISPR-Cas9 knockout cells revealed that virion-associated HBV DNA is sensed by the cGAS/STING pathway, while MAVS is dispensable, ruling out the possible sensing of HBV DNA by MDA5 suggested by Lu et al. [3] Our results confirm and extend earlier studies showing the role of the cGAS/STING pathway in HBV DNA sensing [11,18]. In addition, we demonstrated that the other HBV DNA species tested, HBV replication intermediates, are immunostimulatory as well. Although we primarily used myeloid cells as a model to study the immunostimulatory potential of viral nucleic acids in cells expressing most of the PRRs, a direct role for these cells in the innate response to HBV in vivo has been suggested [8,9,10]. The hypothesis that Kupffer cells or dendritic cells may engulf viral particles or viral debris and respond to them deserves further investigation. Our data show that myeloid cells can mount an innate immune response against HBV DNAs when actively introduced into cell cytoplasm by transfection. However, so far, our preliminary experiments have indicated no innate immune response of THP-1, MDDCs or Kupffer cells to inoculation with HBV particles, neither to naked HBV DNA in the absence of transfection reagent and therefore do not allow us to confirm this hypothesis.

In hepatocytes, the activity of the cGAS/STING pathway is controversial. Several publications have indicated low expression and functionality of these factors in hepatocytes [9,14,19], on the contrary, Verrier et al. [11] proposed that both proteins are expressed and able to sense foreign DNA. Here we confirm that STING is expressed in hepatocytes including PHH but show that its expression is reduced compared to immune cells. cGAS RNA was detected in PHH, but was under the detection limit in HepG2-derived cell lines. Nevertheless, we show that PHH and HepG2-hNTCP are able to sense HBV DNA when transfected in sufficient amounts (Figure 4). Using THP-1 KO cell lines as a model to dissect the possible sensing pathway(s) involved (Figure 2), we demonstrated that HT-DNA and HBV DNA sensing was totally abrogated by cGAS or STING KO, demonstrating that the cGAS/STING pathway is required for HBV DNA sensing and that other possible DNA sensors expressed in THP1 are not sufficient to sense these DNA species in the absence of cGAS or STING. However, we do not exclude that other DNA sensors such as IFI16, which was recently suggested to sense HBV cccDNA in the nucleus of hepatocytes [63], might act as cofactors of the cGAS/STING pathway. These data suggest that in hepatocytes, the cGAS/STING pathway is very likely responsible for the observed sensing of virion-associated HBV DNA, even if cGAS and STING RNA expression is low or below the detection limit of our assay. Furthermore, we demonstrated that the efficiency of the innate response to transfected DNA in hepatocytes is not optimal and can be improved by overexpressing cGAS and STING (Figure 4B). In summary, we propose a central role of the cGAS/STING pathway in HBV DNA sensing. We demonstrated that, in hepatocytes, this pathway is expressed at a relatively low level but is sufficient for sensing high amounts of naked HBV DNA from particles. 

The possible inhibition of innate immune pathways by HBV as a way to escape the interferon response is still a matter of debate. The viral regulatory protein HBx has been described to interfere with MAVS [21,27] leading to a reduced innate immune response to different stimuli in cells transfected with plasmids encoding HBx or the HBV genome. Here we observed no innate immune response in PHH infected with an HBV X- mutant (Figure 5B,C), suggesting that HBx is not responsible for the lack of innate immune response to HBV in infected hepatocytes. Although we cannot exclude that a higher MOI of HBV X- may have led to an innate immune response in PHH, we further showed that MAVS is not involved in sensing HBV nucleic acids, while cGAS and STING are (Figure 2). This confirms the hypothesis that HBx is not responsible for the absence of innate response to HBV infection in hepatocytes. 

Several publications have proposed that HBV does not inhibit innate immune responses. Many of them focused on RNA-sensing pathways [9,31,32], while the DNA-sensing pathway has been less investigated and is still a matter of debate. Liu et al. suggested that HBV replication dampened STING-mediated innate signaling [30]. In addition, Verrier et al. [11] proposed that HBV infection decreases the expression levels of STING, cGAS and TBK1. On the contrary, Guo et al. [12] reported that an induction of HBV replication in HepAD38 cells did not affect the response to foreign DNA. Here, we demonstrated that IRF3 is not phosphorylated in HBV-infected hepatocytes (Figure 5). This indicates that, if the absence of activation of the DNA-sensing pathway following HBV infection is due to an active inhibition by the virus, this viral inhibition would target a step upstream of IRF3 phosphorylation. We then demonstrated that HBV infection does not alter IRF3 nuclear translocation in response to HT-DNA transfection in HepG2-hNTCP overexpressing cGAS and STING (Figure 6). In addition, we observed no significant change in cGAS and STING expression levels in PHH infected with a high MOI of HBV (Appendix A). Altogether, our data point towards an absence of inhibition of the cGAS-STING DNA-sensing pathway by HBV.

In the absence of active inhibition, HBV evasion from the DNA-sensing pathway has to be passive. Since we observed that hepatocytes can sense and respond to naked HBV DNA, we hypothesize that a shielding of viral rcDNA and replication intermediates in the viral capsids may be the reason why HBV DNA is not sensed upon infection. This hypothesis has been suggested in an immortalized mouse hepatocyte cell line model where HBV capsids are destabilized [64]. Nevertheless, it is difficult to accurately quantify and compare the copy numbers of HBV DNA in infected hepatocytes and in our hepatocyte transfection assay. Therefore we cannot exclude that upon infection, the levels of intracellular HBV DNA are not sufficient to activate the cGAS/STING pathway. Both hypotheses may coexist and ensure that the virus is not detected under conditions where small amounts of viral DNA are released in the cytoplasm from defective capsids.

During the preparation of this manuscript, a report was published suggesting that HBV stimulates the TLR-2 pathway leading to NF-κB activation and production of pro-inflammatory cytokines [7]. The authors suggested that a receptor-binding mechanism might be involved in this response. This finding is not contradictory with our results since we demonstrated that upon HBV infection, viral nucleic acids were not detected in hepatocytes, but we cannot exclude that another viral factor might activate a pro-inflammatory response.

## 5. Conclusions

In conclusion, we showed for the first time that HBV RNAs are not immunostimulatory. Furthermore, we indicated that different forms of HBV DNA can be sensed by the cGAS/STING pathway. This pathway is expressed at a low level but is active in hepatocytes and able to respond to naked HBV DNA. However, upon infection, the virus escapes this response, apparently without actively inhibiting the DNA-sensing pathway. The shielding of the viral DNA in the capsids combined to the low functionality of the cGAS/STING pathway likely allow HBV to escape cGAS/STING sensing.

## Figures and Tables

**Figure 1 viruses-12-00592-f001:**
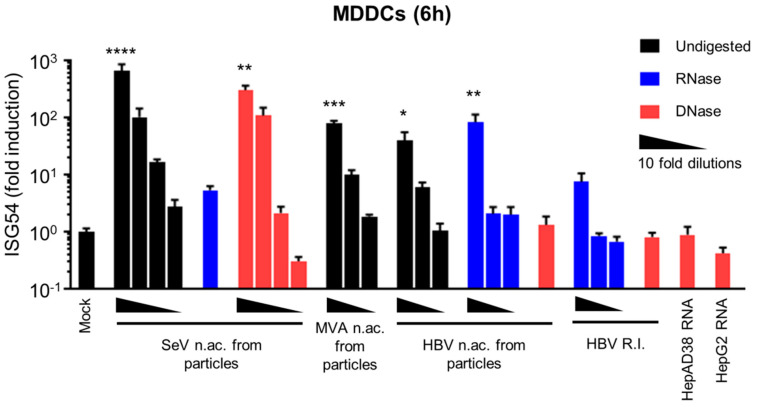
HBV DNAs but not RNAs are immunostimulatory. Total nucleic acids (n.ac.) were extracted from SeV, MVA-gfp or HBV particles. HBV DNA replication intermediates (HBV R.I.) were extracted from HepAD38 cytoplasm. Total RNAs were extracted from HepAD38 or HepG2 cells. When indicated, the nucleic acids were digested with RNAse (blue) or DNAse (red; black bars: undigested). Human MDDCs were transfected with 10-fold dilutions of the nucleic acids (quantified in Table 3). ISG54 mRNA level was quantified by RT-qPCR 6 h post-transfection. Average and SEM of three technical replicates of at least two donors are shown. Levels of significance over mock transfection: **** *p* < 0.0001, *** *p* < 0.001, ** *p* < 0.01 and * *p* < 0.05 (Kruskal–Wallis test with Dunn’s multiple comparisons test).

**Figure 2 viruses-12-00592-f002:**
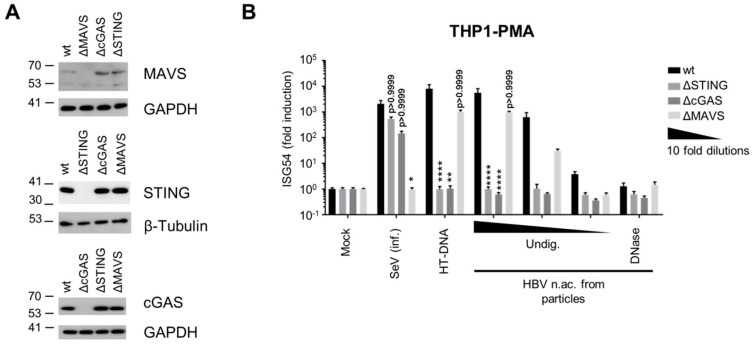
HBV DNA is sensed by the cGAS/STING pathway. (**A**) Genome editing of THP1 CRISPR/Cas9 KO for STING, cGAS or MAVS was controlled by Western Blot. (**B**) PMA-differentiated WT or KO THP1 cells were infected with SeV, transfected with HT-DNA (5 pg/cell) or with serial dilutions of undigested (Undig.) or DNAse-digested HBV nucleic acids (n.ac.) from particles. ISG54 mRNA fold induction to the mock was determined by RT-qPCR 24 h post-transfection. Average and SEM of three independent experiments in triplicates are shown. **** *p* < 0.0001, ** *p* < 0.01 and * *p* < 0.05 (Kruskal–Wallis test with Dunn’s test; for each stimulus, the knock-out cells are compared to the WT cells).

**Figure 3 viruses-12-00592-f003:**
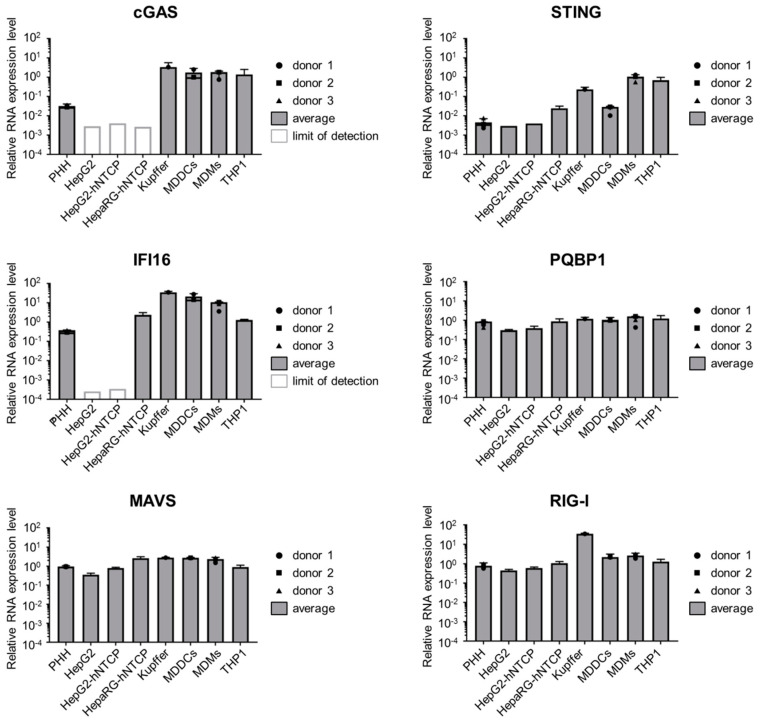
Hepatocytes express a low level of components of the DNA-sensing pathway. cGAS, STING, IFI16, PQBP1, MAVS and RIG-I mRNAs were quantified by RT-qPCR in the indicated cell types and normalized to the geometric mean of RPL13A and TBP mRNAs. For primary cells, the average and SEM of technical triplicates from 3 (PHH and MDMs), 2 (MDDCs) or 1 (Kupffer cells) donors are shown.

**Figure 4 viruses-12-00592-f004:**
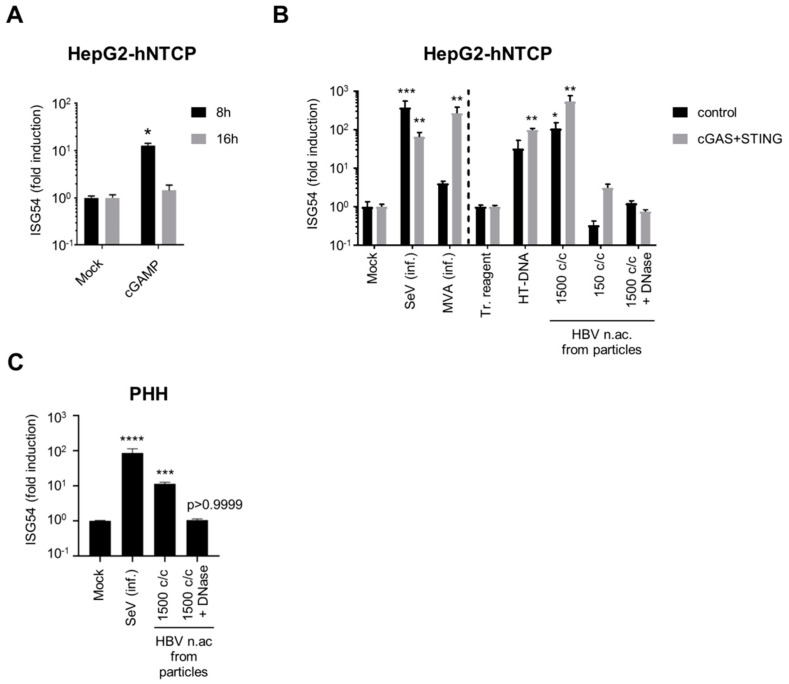
Hepatocytes respond to foreign DNA. HepG2-hNTCP (**A**), HepG2-hNTCP-control or -cGAS/STING (**B**) or PHH (**C**) were stimulated with cGAMP (A), infected (inf.) with SeV or MVA-gfp, or transfected with HT-DNA or HBV nucleic acids (c/c: copies/cells) (**B**,**C**). ISG54 mRNA fold inductions to mock (**A**,**B** left part, **C**) or to transfection reagent-treated cells (tr. reagent) (**B**, right part) was determined by RT-qPCR. Average and SEM of 4 (**A**), 2 (**B**) independent experiments in triplicates or of 4 donors in duplicates (**C**). Levels of significance compared to respective controls (**A**,**B** left part, **C**: mock; **B**, right part: transfection reagent-treated cells): **** *p* < 0.0001, *** *p* < 0.001, ** *p* < 0.01, * *p* < 0.05 (2-way ANOVA with Sidak’s multiple comparisons (**A**); Kruskal–Wallis with Dunn’s test (**B**,**C**)).

**Figure 5 viruses-12-00592-f005:**
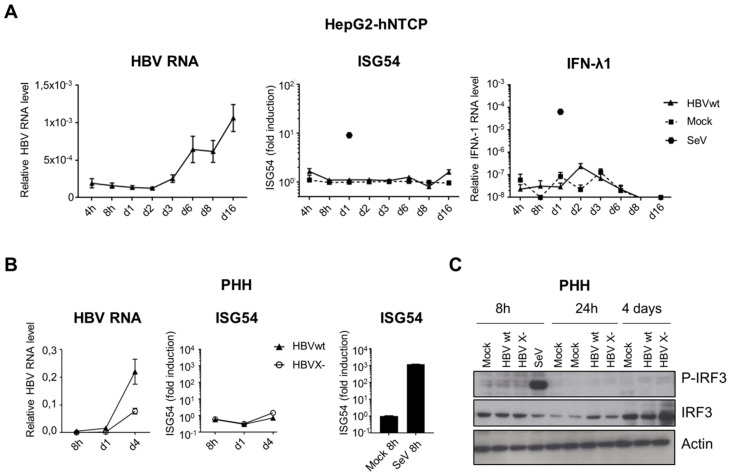
HBVwt or HBV X- infection do not activate nucleic acid-sensing pathways in hepatocytes. HepG2-hNTCP (**A**) or PHH (**B**,**C**) were infected with HBVwt (MOI 100), HBV X- (MOI 100) or SeV (MOI 0,13). HBV, ISG54 or IFN-λ1 RNAs were quantified by RT-qPCR (**A**,**B**). Relative expression levels to the reference gene RPL13A were calculated. For ISG54, fold induction to the respective mock-control is shown for each time point. Average and standard error of the mean (SEM) of three independent (HepG2-hNTCP) or 1 (PHH) experiment in technical triplicates are shown. (**C**) Phospho- or total IRF3 protein levels were analyzed by Western blot in HBV-infected PHH (HBVwt or HBV X-, MOI 100).

**Figure 6 viruses-12-00592-f006:**
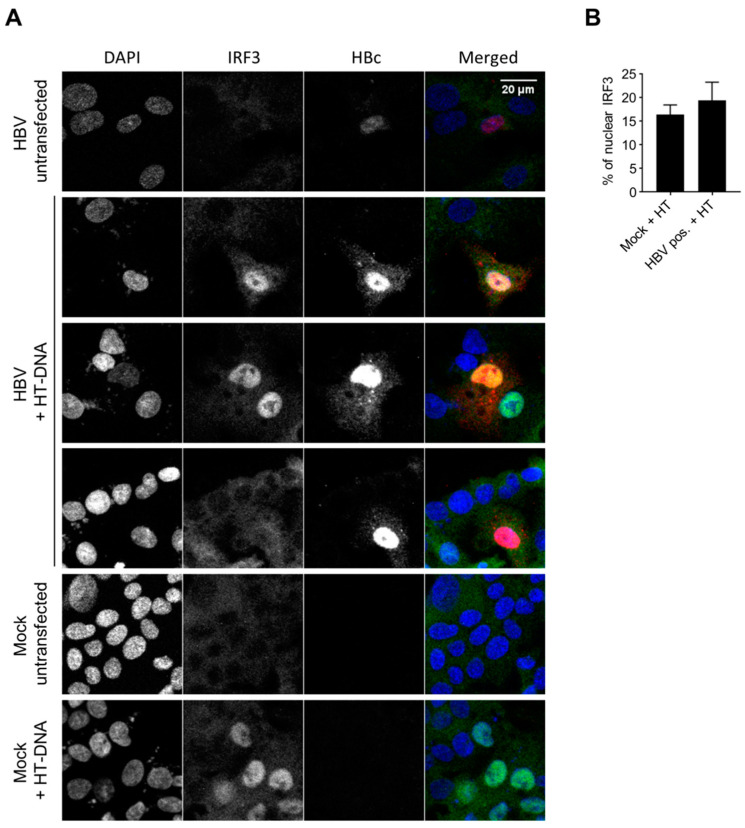
HBV does not inhibit the innate response to foreign DNA. HepG2-hNTCP-cGAS-STING were infected or not with HBVwt (MOI 10000). Seven days post-infection (dpi), the DNA-sensing pathway was stimulated with HT-DNA. Sixteen hours post-HT-DNA-transfection, IRF3 (green), HBc (red) and DNA (blue) were stained. (**A**) Representative images. (**B**) Percentage of nuclear IRF3 counted in HT-DNA transfected cells. For HBV-infected cells, the analysis is restricted to the HBc-positive cells (productive infection). Average and SEM of two independent experiments in duplicates are shown. At least 67 cells per condition and replicate were counted (total: 1153 mock-infected cells; 333 HBc-positive cells).

**Table 1 viruses-12-00592-t001:** Cell lines, suppliers and authentication methods.

Name	Citation	Supplier	Authentication Test Method
HepG2	Figure 3	Gift from Prof S.Urban, Heidelberg, Germany	Cell line authentication performed by Eurofins Genomics, using Applied BiosystemsTM AmpFLSTRTM IdentifilerTM Plus PCR Amplification Kit with 16 markers
HepG2-hNTCP	Figures 3–6	Gift from Prof S.Urban, Heidelberg, Germany [33]	Cell line authentication performed by Eurofins Genomics, using Applied BiosystemsTM AmpFLSTRTM IdentifilerTM Plus PCR Amplification Kit with 16 markers
HepaRG-hNTCP	Figure 3	Gift from Prof S.Urban, Heidelberg, Germany	Multiplexion cell contamination assay (Heidelberg, Germany) as described [39]
HepAD38	Figure 1 (for HBV RNAs and replication intermediate extraction)	Gift from Prof E Hild, Langen, Germany	Release of HBV particles (qPCR + infectivity test)
THP1	Figures 2 and 3	Gift from Prof V. Hornung, Munich, Germany	Cell line authentication by Eurofins Genomics on 24.03.2017
THP1 ΔSTING	Figure 2	Gift from Prof V. Hornung, Munich, Germany	Western blot
THP1 ΔcGAS	Figure 2	Gift from Prof V. Hornung, Munich, Germany	Western blot
THP1 ΔMAVS	Figure 2	Gift from Prof V. Hornung, Munich, Germany	Western blot

**Table 2 viruses-12-00592-t002:** Oligonucleotides.

Name	Sequence (5′–3′)	Supplier	Use
HBV_RC_F	CACTCTATGGAAGGCGGGTA	Eurofins Genomics	qPCR for quantification of HBV stocks
HBV_RC_R	TGCTCCAGCTCCTACCTTGT	Eurofins Genomics	qPCR for quantification of HBV stocks
HBV RNA 3 F	GCTTTCACTTTCTCGCCAAC	Eurofins Genomics	RT-qPCR
HBV RNA 3 R	GAGTTCCGCAGTATGGATCG	Eurofins Genomics	RT-qPCR
ISG54_F	GGTGGCAGAAGAGGAAGATT	Eurofins Genomics	RT-qPCR
ISG54_R	TAGGCCAGTAGGTTGCACAT	Eurofins Genomics	RT-qPCR
IFNλ-1_F	CGCCTTGGAAGAGTCACTCA	Eurofins Genomics	RT-qPCR
IFNλ-1_R	GAAGCCTCAGGTCCCAATTC	Eurofins Genomics	RT-qPCR
cGAS_F	CAAGAAGGCCTGCGCATTCA	Eurofins Genomics	RT-qPCR
cGAS_R	GAGAAGGATAGCCGCCATGT	Eurofins Genomics	RT-qPCR
STING_F	GATATCTGCGGCTGATCCTG	Eurofins Genomics	RT-qPCR
STING_R	GCTGTAAACCCGATCCTTGA	Eurofins Genomics	RT-qPCR
IFI16_F	CGCTTGAAGACCTGGCTGAA	Eurofins Genomics	RT-qPCR
IFI16_R	TGACAGTGCTGCTTGTGGAG	Eurofins Genomics	RT-qPCR
PQBP1_F	TCTGGAGCCTGAACCAGAGGAA	Eurofins Genomics	RT-qPCR
PQBP1_R	TCCAACCTGGTGGCCTCGTAGT	Eurofins Genomics	RT-qPCR
RIG-I_F	CCTACCTACATCCTGAGCTACAT	Eurofins Genomics	RT-qPCR
RIG-I_R	TCTAGGGCATCCAAAAAGCCA	Eurofins Genomics	RT-qPCR
MAVS_F	GGTGCTCACCAAGGTGTCTG	Eurofins Genomics	RT-qPCR
MAVS_R	AGGAGGTGCTGGCACTGATG	Eurofins Genomics	RT-qPCR
MDA5_F	AGAGTGGCTGTTTACATTGCC	Eurofins Genomics	RT-qPCR
MDA5_R	GCTGTTCAACTAGCAGTACCTT	Eurofins Genomics	RT-qPCR
TLR3_F	ACCTCCAGCACAATGAGCTA	Eurofins Genomics	RT-qPCR
TLR3_R	TCCAGCTGAACCTGAGTTCC	Eurofins Genomics	RT-qPCR
RPL13A_F	CCT GGA GGA GAA GAG GAA AGA GA	Eurofins Genomics	RT-qPCR
RPL13A_R	TTG AGG ACC TCT GTG TAT TTG TCA A	Eurofins Genomics	RT-qPCR
TBP_F	GGAGCTGTGATGTGAAGT	Eurofins Genomics	RT-qPCR
TBP_R	TACGTCITCTTCCTGAATCC	Eurofins Genomics	RT-qPCR
HBV_quant_F	TGTCAACACTAATATGGGCCTAA	Eurofins Genomics	quantification of HBV RNA and DNA
HBV_quant_R	AGGGGCATTTGGTGGTCTAT	Eurofins Genomics	quantification of HBV RNA and DNA
HBV_st	TGTCAACACTAATATGGGCCTAAAGTTCAGGCAACTCTTGTGGTTTCACATTTCTTGTCTCACTTTTGGAAGAGAAACAGTTATAGAGTATTTGGTGTCTTTCGGAGTGTGGATTCGCACTCCTCCAGCTTATAGACCACCAAATGCCCCT	Integrated DNA Technologies, BvBA	standard for quantification of HBV RNA and DNA
MVAgfp_F	AGCACGACTTCTTCAAGTCC	Eurofins Genomics	quantification of MVAgfp DNA
MVAgfp_R	GTTGTAGTTGTACTCCAGCTTG	Eurofins Genomics	quantification of MVAgfp DNA
MVAgfp_st	AGCACGACTTCTTCAAGTCCGCCATGCCCGAAGGCTACGTCCAGGAGCGCACCATCTTCTTCAAGGACGACGGCAACTACAAGACCCGCGCCGAGGTGAAGTTCGAGGGCGACACCCTGGTGAACCGCATCGAGCTGAAGGGCATCGACTTCAAGGAGGACGGCAACATCCTGGGGCACAAGCTGGAGTACAACTACAAC	Integrated DNA Technologies, BvBA	standard for quantification of MVAgfp DNA
SeV_F	TTCATTATCATCCCGTGAGA	Eurofins Genomics	quantification of SeV RNA
SeV_R	CCAGTGATCCATCATCAATC	Eurofins Genomics	quantification of SeV RNA
SeV_st	TTCATTATCATCCCGTGAGATCAGGAACCTGAGGGTTATCACAAAAACTTTATTAGACAGGTTTGAGGATATTATACATAGTATAACGTATAGATTCCTCACCAAAGAGATAAAGATTTTGATGAAGATTTTAGGGGCAGTCAAGATGTTCGGGGCCAGGCAAAATGAATACACGACCGTGATTGATGATGGATCACTGG	Integrated DNA Technologies, BvBA	standard for quantification of SeV RNA

(Eurofins Genomics, Ebersberg, Germany; Integrated DNA Technologies, BvBA, Coralville, IA, USA).

**Table 3 viruses-12-00592-t003:** Quantification of viral nucleic acids used for MDDC transfection experiments.

	SeV n.ac. from Particles	MVA n.ac. from Particles	HBV n.ac. from Particles	HepaD38 RNA	HBV R.I.
Viral DNA (copies/cell)		5.0E+03	5.0E+03		5.0E+03
Viral RNA (cDNA-equivalent copies/cell)	2.8E+01		2.3E+01	1.3E+04	

(n.ac: nucleic acids; R.I.: replication intermediates).

**Table 4 viruses-12-00592-t004:** Antibodies used for Western blots.

Name	Supplier	Cat No.	Clone No.
Anti phospho S386-IRF3 (rabbit)	Abcam	ab76493	EPR2346
Anti IRF3 (rabbit)	Epitomics	2241-1	EP2419Y
Anti beta-actin (mouse)	Sigma-Aldrich	A5441	AC-15
Anti beta-tubulin (mouse)	Sigma-Aldrich	075K4875	TUB 2.1
Anti GAPDH (rabbit)	Cell Signaling Technology	2118	14C10
Anti MAVS (rabbit)	Abcam	Ab25084	
Anti STING (rabbit)	Cell Signaling Technology	13647	D2P2F
Anti cGAS (rabbit)	Cell Signaling Technology	15102	D1D3G
Anti-mouse HRP	Cell Signaling Technology	7076S	
Anti-rabbit HRP	Cell Signaling Technology	7074S	

(Abcam, Cambridge, United Kingdom; Epitomics, Burlingame, CA, USA; Sigma-Aldrich, St. Louis, MO, USA; Cell Signaling Technology, Danvers, MA, USA).

**Table 5 viruses-12-00592-t005:** Antibodies used for immunofluorescence.

Name	Citation	Supplier	Cat No.	Clone No.
Anti IRF3 (rabbit)	IRF3 translocation assay, Figure 6	Cell Signaling Technology	11904S	D6I4C
anti HBc (mouse)	IRF3 translocation assay, Figure 6	gift from Prof. S. Urban, Heidelberg		M312
anti-rabbit Alexa Fluor 488	IRF3 translocation assay, Figure 6	Invitrogen	A11008	
anti-mouse Alexa Fluor 555	IRF3 translocation assay, Figure 6	Thermo Fischer Scientific	A21422	
Anti-HBc (rabbit)	Appendix A	Dako, Agilent	B0586	
Goat anti-rabbit IgG(H+L), Alexa Fluor 546	Appendix A	Thermo Fischer Scientific	A-11010	

(Cell Signaling Technology, Danvers, Massachusetts, USA; Invitrogen, Carlsbad, CA, USA; Thermo Fischer Scientific, Waltham, MA, USA; Dako, Agilent, Santa Clara, CA, USA).

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
