# Peer review of "Hepatitis B Virus DNA is a Substrate for the cGAS/STING Pathway but is not Sensed in Infected Hepatocytes"

_viruses, 2020, doi:10.3390/v12060592_

Round 1
Reviewer 1 Report
The paper by Lise Lauterbach-Riviere et al. has investigated in vitro if HBV nucleic acids are sensed by immune cells and hepatocytes via the cGAS/STING receptor pathway. The main results demonstrate that (1) HBV pgRNA and mRNAs are not immunostimulatory in MDDCs and THP1; (2) HBV rcDNA within secreted virions is immunostimulatory in MDDCs, THP1, HepG2-hNTCP, and PHHs and sensed by cGAS/STING; (3) HBV infection in HepG2-hNTCP and PHHs is not inducing an innate immune response via cGAS/STING under the selected experimental conditions, and (4) HBV infection is not actively inhibiting this cytosolic viral DNA sensing receptor pathway in HepG2-hNTCP in response to foreign DNA. The conclusions are that HBV during infection of hepatocytes evades cytoplasmatic recognition by cGAS/STING possibly through shielding of rcDNA within the nucleocapsid and by producing RNAs that are not immunostimulatory.
Overall, the experiments were well planned and carefully executed. The results are well described but sometimes hard to follow in the manuscript. Nevertheless, it is an important study, because it further dissects the mechanisms by which HBV evades recognition via the innate immune response, especially those involving cytosolic viral DNA sensing receptors in hepatocytes.
Major comments:
- Figure 1: Without defining the cut-off value for ISG54 induction (i.e., the value that was considered a positive response to DNA/RNA stimulation) and absence of significant induction over mock control, it is not clear why the authors considered HBV RIs to be immunostimulatory in MDCCs. Also, the statement on line 285 that the ISG54 response was slightly weaker appears incorrect as the difference to rcDNA was about 10-fold or 1 log. This statement should be modified and other statements that HBV RIs were immunostimulatory should be tuned down in the manuscript.
- Figure 5: Based on the relative HBV RNA levels, it appears that HBV infection in HepG2-hNTCP and PHHs was productive but that HBV RIs were not sensed via cGAS/STING. However, it is not clear if the concentrations of HBV RIs at days 16 or 4 in both cells were broadly comparable to those used in Figure 1 (i.e., 5.0E+03) that induced ISG54 expression in MDCCs. In addition, HBVX- RNA level in PHHs at day 4 was about 5-fold lower than the HBVwt RNA level but appeared to result in an accumulation of IRF3 protein. Thus, it is not clear if a HBVX- RNA level comparable to the HBVwt RNA level would have led to (detectable) IRF3 phosphorylation by Western blot. An important control experiment with HBV rcDNA that was immunostimulatory in both cells (Figure 4) and that would be expected to result in IRF3 phosphorylation after sensing by cGAS/STING has not been performed/shown. Thus, the statement on lines 391 to 394 and comparable statements should be modified.
Minor comments:
- Lines 96, 98, and 101: Please describe the cell lines HepAD38, HepaRG-hNTCP, and THP1 in more detail (i.e., comparable to HepG2-hNTCP, HepG2.2.15, and HepG2 H1.3deltaX cell lines).
- Table 1: Should read HepAD38 instead of HepaD38
- Line 110: Please describe in some detail the patients from which PHHs were isolated; i.e., reason for hepatectomy and/or HBV and any other viral status.
- Lines 155 and 163: Please describe acronyms the first time they are used in the manuscript (i.e., TATBP and HT-DNA)
- Figure 1: Please explicitly state the ISG54 fold induction value that was considered a positive response to DNA/RNA stimulation. It appears that a 10-fold change was used as the cut-off, although this was insignificant to mock transfection. Please explain in the figure legend what the levels of significance mean (i.e., significant ISG54 induction over mock transfection).
- Lines 357 and 358: The statement “while the infected samples were compared to an uninfected/untransfected control (mock)” is not clear. Please modify and state what are the infected samples.
- Figure 5: Similar comment as for Figure 1: Without defining the cut-off value for ISG54 induction and absence of significant induction over mock control, it is not clear why HT-DNA and MVA-gf were considered immunostimulatory in HepG2-hNTCP. Please explain in the figure legend what the levels of significance mean.
- Figure 6: Could the authors please comment why HepG2-hNTCP rather than PHHs were used for this experiment. PHHs express cGAS and IFI16 while gene transcription in HepG2-hNTCP is undetectable. The concern is that an (artificial) overexpression of cGAS and STING in HepG2-hNTCP may always result in a strong IRF3 nuclear translocation following transfection of HT-DNA, and without detectable interference by HBV. Such interference may be detectable (or not) under more “natural” conditions in PHHs.
Author Response
We would like to express our appreciation for the helpful and constructive feedback from the reviewers. We revised our manuscript accordingly to all issues that have been raised by the reviewers. In our responses, we referred to line numberings of the manuscript version in tracked-changes.
Reviewer’s Comment:
The paper by Lise Lauterbach-Riviere et al. has investigated in vitro if HBV nucleic acids are sensed by immune cells and hepatocytes via the cGAS/STING receptor pathway. The main results demonstrate that (1) HBV pgRNA and mRNAs are not immunostimulatory in MDDCs and THP1; (2) HBV rcDNA within secreted virions is immunostimulatory in MDDCs, THP1, HepG2-hNTCP, and PHHs and sensed by cGAS/STING; (3) HBV infection in HepG2-hNTCP and PHHs is not inducing an innate immune response via cGAS/STING under the selected experimental conditions, and (4) HBV infection is not actively inhibiting this cytosolic viral DNA sensing receptor pathway in HepG2-hNTCP in response to foreign DNA. The conclusions are that HBV during infection of hepatocytes evades cytoplasmatic recognition by cGAS/STING possibly through shielding of rcDNA within the nucleocapsid and by producing RNAs that are not immunostimulatory.
Overall, the experiments were well planned and carefully executed. The results are well described but sometimes hard to follow in the manuscript. Nevertheless, it is an important study, because it further dissects the mechanisms by which HBV evades recognition via the innate immune response, especially those involving cytosolic viral DNA sensing receptors in hepatocytes.
We thank the reviewer for appreciating our findings. We feel that the revisions based on suggestions of the reviewer have significantly strengthened this manuscript and substantiated the conclusions of the study.
Major comments:
- Figure 1: Without defining the cut-off value for ISG54 induction (i.e., the value that was considered a positive response to DNA/RNA stimulation) and absence of significant induction over mock control, it is not clear why the authors considered HBV RIs to be immunostimulatory in MDCCs. Also, the statement on line 285 that the ISG54 response was slightly weaker appears incorrect as the difference to rcDNA was about 10-fold or 1 log. This statement should be modified and other statements that HBV RIs were immunostimulatory should be tuned down in the manuscript.
We thank the reviewer for carefully analyzing this figure and we recognize that HBV replication intermediates do not show a significant induction over the mock control at the time point that was shown in Fig. 1 (6h post transfection). However, we have data at a later time point (24h) for 2 out of the 3 MDDC donors used in Fig. 1.
Therefore we now globally analyzed the two time points (6h + 24h) by performing a 2 way ANOVA with Tukey‘s multiple comparison (“2 way ANOVA main raw effect“ using Prism 8.4.2) and this test shows a significant ISG54 induction compared to the mock for the undiluted HBV R.I. For consistency, and in order to exclude that HBV RNAs may induce an ISG54 peak at 24h hours, HepAD38 RNA and HepG2 RNA have also been analyzed in the same way. These data are now presented in the new Figure S1. We therefore conclude that HBV R.I. are immunostimulatory and confirm that HBV RNAs are not. Since the provided data in new Figure S1 show that HBV R.I. are significantly immunostimulatory when the 6h and 24h time points are analyzed, we consider that this result is still interesting and worth mentioning. We rewrote the whole paragraph (line 294-320)
We recognize that using a cut-off value would be arbitrary when not supported by statistical significance, therefore we now removed in the text all sentences considering as immunostimulatory samples that induce a more than 10-fold ISG54 induction, if this was not statistically significant. Therefore we do not mention anymore the “limits of detection” for SeV RNAs or MVA DNAs and only compare HBV RNAs to the highest concentration of SeV RNAs, which induces a significant ISG54 induction (lines 265-272; 288-293; 314-320). We therefore tuned down the conclusion that particle-associated HBV RNAs are not immunostimulatory per se (since the amount is too low to conclude). We only conclude that they do not account for the ISG54 induction observed with HBV nucleic acids from particles (lines 288-293 and 322-323; 337-338 in results and lines 498-500in discussion). However we kept the conclusion that HBV RNAs extracted from HepAD38 are not immunostimulatory, since we used 460 times more HBV RNAs (1,3E+4 cDNA-equivalent copies/cells) than the amount of SeV RNAs that significantly induces ISG54 (28 cDNA-equivalent copies/cells).
- Figure 5: Based on the relative HBV RNA levels, it appears that HBV infection in HepG2-hNTCP and PHHs was productive but that HBV RIs were not sensed via cGAS/STING. However, it is not clear if the concentrations of HBV RIs at days 16 or 4 in both cells were broadly comparable to those used in Figure 1 (i.e., 5.0E+03) that induced ISG54 expression in MDCCs. In addition, HBVX- RNA level in PHHs at day 4 was about 5-fold lower than the HBVwt RNA level but appeared to result in an accumulation of IRF3 protein. Thus, it is not clear if a HBVX- RNA level comparable to the HBVwt RNA level would have led to (detectable) IRF3 phosphorylation by Western blot. An important control experiment with HBV rcDNA that was immunostimulatory in both cells (Figure 4) and that would be expected to result in IRF3 phosphorylation after sensing by cGAS/STING has not been performed/shown. Thus, the statement on lines 391 to 394 and comparable statements should be modified.
We agree that it is difficult to compare the transfection experiments with the infection experiments, since we do not know the amount of viral DNA species that are present in the cells in both cases. It was unfortunately impossible to accurately determine it in case of transfection, since we don’t know the transfection efficiency. PCR analysis of the transfected cells have a very high risk of being contaminated with DNA that is sticking to the outside of the cells, or partially internalized (for example in phagosomes), that does not account for an innate response within the IRF3/ISG54 pathway (indeed we observed that in the absence of transfection reagent, viral DNA does not induce an innate response in MDDCs).
We therefore modified our conclusions and explained that we cannot exclude that in infected hepatocytes, the level of intracellular HBV DNA is not sufficient to activate the cGAS/STING pathway (Results line 392-393, 431-433 and discussion lines 484, 529, 544-546, 576-581, 595-596 + abstract lines 30-31).
We did not perform PHH infections with higher MOIs of HBV X-, therefore we cannot rule-out that higher levels of HBV X- replication may lead to detectable IRF3 phosphorylation. We now modified the text accordingly (Line 424-425 and discussion 551-556). However, HBx was proposed to interfere with MAVS (1,2), while we showed that MAVS is not involved in sensing HBV nucleic acids (Figure 1 shows that HBV RNAs are not immunostimulatory and Figure 3 shows that MAVS does not sense HBV DNAs). Our results therefore do not support a role of HBx in inhibiting the innate response to HBV).
We did not formally show that HBV DNA transfection in hepatocytes induces IRF3 phosphorylation. However, we showed that HBV DNA is sensed by the cGAS/STING pathway and induces ISG54. To our knowledge, there has been no reports on ISG54 induction upon cGAS/cGAMP activation in the absence of IRF3 activation (3,4). Intracellular cytosolic DNA and RNA sensors are thought to mainly activate IRF3 (5,6). ISG54 is considered a direct downstream transcriptional target of IRF-3 (7,8). We recognize that other IRF family members may play a role besides IRF3. For instance, IRF3 and IRF7, are the principal mediators of IFN induction, acting downstream of cytosolic RNA and DNA receptors and the TLRs ( 9). Additionally, IRF5 has been shown to play an immune protective role against infection by various RNA viruses, however, in contrast to IRF3 and IRF7, the role of IRF5 in regulating antiviral defences and viral infection is not well understood (10). IRF3 is ubiquitously expressed, whereas IRF7 is expressed only at very low levels, except in plasmacytoid DCs where it is relatively abundant. IRF7 forms a heterodimer with IRF3 to induce the production of IFNα4 and IFNβ in the early phase of the response. The initially induced type I IFN then, in turn, activates the expression of IRF7, which participates in the induction of most IFNα subtypes, thus functioning as a key mediator of the type I IFN amplification loop in the later phase of the response. In summary, even if feed-forward loops may involve other transcription factors, to our knowledge, the first signalling events following sensing by cGAS, which is what we want to study, always involve IRF3. To our knowledge, studies on cytosolic DNA responses in hepatocytes without involvement of IRF3 has not yet been published, however we formally cannot exclude the possibility of the involvement of a non-canonical transcription factor.
Minor comments:
- Lines 96, 98, and 101: Please describe the cell lines HepAD38, HepaRG-hNTCP, and THP1 in more detail (i.e., comparable to HepG2-hNTCP, HepG2.2.15, and HepG2 H1.3deltaX cell lines).
We now added the description and references for the cell lines HepAD38, HepaRG-hNTCP, and THP1. We also noticed and corrected a mistake in the reference for HepG2.2.15 (former reference was Ladner et al. 1997; new reference is Sell et al. 1987) and for the description of HepG2-hNTCP (lines 91-93): “have been stably transduced with a lentiviral vector expressing HBV receptor, the sodium-taurocholate cotransporting polypeptide (NTCP)” instead of “have been stably transfected with HBV receptor, the sodium-taurocholate cotransporting polypeptide (NTCP)”. We apologize for these mistakes.
- Table 1: Should read HepAD38 instead of HepaD38
This mistake has been corrected.
- Line 110: Please describe in some detail the patients from which PHHs were isolated; i.e., reason for hepatectomy and/or HBV and any other viral status.
We now added the reason for hepatectomy (hepatocellular carcinoma or severe cirrhosis) as well as the viral status (negative for HBV, HCV and HIV) of the patients from which PHHs were isolated. We also added details about the PHH purchased from Thermo Fisher Scientific (“These cryopreserved PHH were isolated from dead patients who were tested negative for HBV, HCV, HIV1 and 2, HTLV 1 and 2, CMV”, line 122-123)
- Lines 155 and 163: Please describe acronyms the first time they are used in the manuscript (i.e., TATBP and HT-DNA)
We added the description of the acronyms: HT-DNA: herring testis DNA (line 172); TATBP: TATA-box binding protein. The acronym TATBP was replaced by TBP, which is more commonly used (lines 163, Table2).
- Figure 1: Please explicitly state the ISG54 fold induction value that was considered a positive response to DNA/RNA stimulation. It appears that a 10-fold change was used as the cut-off, although this was insignificant to mock transfection. Please explain in the figure legend what the levels of significance mean (i.e., significant ISG54 induction over mock transfection).
As explained in the response to the first major point, we recognize that using a cut-off value would be arbitrary when not supported by statistical significance, therefore we now removed in the text all sentences considering as immunostimulatory samples that induce a more than 10-fold ISG54 induction, if this was not statistically significant. The list of changes is detailed in the response to the first major point. In addition, we explained in the figure legend that the levels of significance are over mock transfection (line 282).
- Lines 357 and 358: The statement “while the infected samples were compared to an uninfected/untransfected control (mock)” is not clear. Please modify and state what are the infected samples.
We now explained what the “infected samples” are and why they have been compared to the uninfected/untransfected control (mock): “while the samples infected with SeV or MVA, which have not been treated with transfection reagent, were compared to an uninfected/untransfected control (mock).” (lines 386-387)
- Figure 5: Similar comment as for Figure 1: Without defining the cut-off value for ISG54 induction and absence of significant induction over mock control, it is not clear why HT-DNA and MVA-gf were considered immunostimulatory in HepG2-hNTCP. Please explain in the figure legend what the levels of significance mean.
We suppose that the reviewer means Figure 4 instead of Figure 5.
Similar to Figure 1, we recognize that using a cut-off would not be accurate if not supported by statistical analysis. We therefore removed in the text all sentences referring to an induction when this was not statistically significant (i.e.: references to HT-DNA and MVA-gfp in HepG2-hNTCP control, line 389-390 in results)
In addition, we noticed and corrected a mistake in the Figure 4B: the stars indicating the levels of significance over mock were missing in the left part of the graph (*** for CTR SeV; ** for cGAS/STING SeV; ** for cGAS/STING MVA).
We explained the levels of significance in the figure legend: “Levels of significance compared to respective controls (A, B left part, C: mock; B, right part: transfection reagent-treated cells):”
- Figure 6: Could the authors please comment why HepG2-hNTCP rather than PHHs were used for this experiment. PHHs express cGAS and IFI16 while gene transcription in HepG2-hNTCP is undetectable. The concern is that an (artificial) overexpression of cGAS and STING in HepG2-hNTCP may always result in a strong IRF3 nuclear translocation following transfection of HT-DNA, and without detectable interference by HBV. Such interference may be detectable (or not) under more “natural” conditions in PHHs.
We agree that using PHH instead of HepG2-hNTCP overexpressing cGAS and STING would be more relevant for this experiment. We tried performing this experiment in PHH, unfortunately the percentage of IRF3 nuclear translocation in our experimental settings was too low to conclude. In fact, detection of IRF3 nuclear localization is always more difficult to detect than ISG54 induction (we observe “only”~16 % of nuclear translocation in HepG2-hNTCP overexpression cGAS and STING, while ISG54 induction over transfection reagent treated control is about 2 log in these cells). This might be due to the fact that IRF3 nuclear localization is transient and that at a given time point, only a fraction of responding cells have IRF3 in the nucleus. To our knowledge, so far, previous experiments investigating the possible inhibition of innate immune response in HBV-infected hepatocytes had been performed with RNA stimulus, but not DNA pathway stimulation (ref 9, 31, 32 of the paper: 11-13). Therefore, we consider that the experiment in Figure 6 provides new information and is worth mentioning, although the cellular system is not optimal.
However, we agree that this point should be mentioned in the text, and now added the following comments and tuned down our conclusion:
Line 31 (abstract) “Finally, our data suggest that HBV infection does not actively inhibit the DNA sensing pathway.” Instead of “Finally, we show that HBV infection does not actively inhibit the DNA sensing pathway.”
Line 87 :” although our data suggest that the virus does not actively suppress it.”
lines 460-464: “This indicates that HBV infection does not inhibit the innate immune response to foreign DNA in HepG2-hNTCP overexpressing cGAS and STING. However, because of the low percentage of IRF3 nuclear localization upon HT-DNA transfection in the absence of cGAS/STING overexpression, we could not perform this experiment in PHH. Therefore we cannot confirm that HBV does not interfere with the DNA sensing pathway in cells expressing a low level of sensors.”
line 487 ” We further demonstrated that HBV does not actively inhibit the innate response to foreign DNA in infected hepatocytes overexpressing cGAS and STING”
lines 567-568: “We then demonstrated that HBV infection does not alter IRF3 nuclear translocation in response to HT-DNA transfection in HepG2-hNTCP overexpressing cGAS and STING (Figure 6).
line 593: “the virus escapes this response, apparently without actively inhibiting the DNA-sensing pathway”
In addition to the changes mentioned above, we added one supplemental figure (Figure S1). We therefore changed the numbering of the other supplemental figures accordingly.
References
- Kumar M, Jung SY, Hodgson AJ, Madden CR, Qin J, Slagle BL. Hepatitis B virus regulatory HBx protein binds to adaptor protein IPS-1 and inhibits the activation of beta interferon. J Virol. 2011;85(2):987-995. doi:10.1128/JVI.01825-10
- Wei C, Ni C, Song T, et al. The hepatitis B virus X protein disrupts innate immunity by downregulating mitochondrial antiviral signaling protein. J Immunol. 2010;185(2):1158-1168. doi:10.4049/jimmunol.0903874
- Wu J, Sun L, Chen X, et al. Cyclic GMP-AMP is an endogenous second messenger in innate immune signaling by cytosolic DNA. Science. 2013;339(6121):826-830. doi:10.1126/science.1229963
- Chen Q, Sun L, Chen ZJ. Regulation and function of the cGAS-STING pathway of cytosolic DNA sensing. Nat Immunol. 2016;17(10):1142-1149. doi:10.1038/ni.3558
- Ablasser A, Hur S. Regulation of cGAS- and RLR-mediated immunity to nucleic acids. Nat Immunol. 2020;21(1):17-29. doi:10.1038/s41590-019-0556-1
- Zhao G, An B, Zhou H, et al. Impairment of the retinoic acid-inducible gene-I-IFN-β signaling pathway in chronic hepatitis B virus infection. Int J Mol Med. 2012;30(6):1498-1504. doi:10.3892/ijmm.2012.1131
- Nakaya T, Sato M, Hata N, et al. Gene induction pathways mediated by distinct IRFs during viral infection. Biochem Biophys Res Commun. 2001;283(5):1150-1156. doi:10.1006/bbrc.2001.4913
- Grandvaux N, Servant MJ, tenOever B, et al. Transcriptional profiling of interferon regulatory factor 3 target genes: direct involvement in the regulation of interferon-stimulated genes. J Virol. 2002;76(11):5532-5539. doi:10.1128/jvi.76.11.5532-5539.2002
- Jefferies CA. Regulating IRFs in IFN Driven Disease. Front Immunol. 2019;10:325. doi:10.3389/fimmu.2019.00325
- Chow KT, Driscoll C, Loo Y-M, Knoll M, Gale M. IRF5 regulates unique subset of genes in dendritic cells during West Nile virus infection. J Leukoc Biol. 2019;105(2):411-425. doi:10.1002/JLB.MA0318-136RRR
- Cheng X, Xia Y, Serti E, et al. Hepatitis B virus evades innate immunity of hepatocytes but activates cytokine production by macrophages. Hepatology. 2017;66(6):1779-1793. doi:10.1002/hep.29348
- Suslov A, Boldanova T, Wang X, Wieland S, Heim MH. Hepatitis B Virus Does Not Interfere With Innate Immune Responses in the Human Liver. Gastroenterology. 2018;154(6):1778-1790. doi:10.1053/j.gastro.2018.01.034
- Mutz P, Metz P, Lempp FA, et al. HBV Bypasses the Innate Immune Response and Does Not Protect HCV From Antiviral Activity of Interferon. Gastroenterology. 2018;154(6):1791-1804.e22. doi:10.1053/j.gastro.2018.01.044
- Donà F, Houseley J. Unexpected DNA loss mediated by the DNA binding activity of ribonuclease A. PLoS ONE. 2014;9(12):e115008. doi:10.1371/journal.pone.0115008
Reviewer 2 Report
HBV is referred to as a stealth virus because of its ability to evade induction of the IRF3-mediated immune response. Several studies have now been performed to understand the mechanisms at play, in particular the relationships between HBV and the two major IFN induction pathways cGAS/STING or RIG-I/MAVS. In this work, the authors rule out any ability of HBV RNA to induce the IRF3-mediated immune response. The experiments are solid and well conducted and make elegant use of immune cells as cellular models including cells deprived of STING, cGAS or MAVS by genome editing, before moving on to hepatocytes. They performed a systematic comparison of the HBV nucleic acids, DNA and RNA, either extracted directly from the viral particles or from the infected cells. The authors confirmed that, in spite of the ability of its DNA to activate the cGAS/STING pathway , HBV cannot activate this pathway upon infection. The authors ruled out an inhibitory effect due to the HbX protein as well as showing that HBV infection does not prevent innate immune response to foreign DNA.
This work provides a solid confirmation of data that have been published recently, i.e, HBV infection cannot induce the IRF3 activation pathway, in spite of the ability of its DNA to activate the cGAS/STING pathway. In addition, this work brings novel results, such as ruling the role of HBV RNA in triggering an immune response, ruling out a role for HBx in controlling the immune response and demonstrating that HBV infection does not prevent induction of the immune response by foreign DNA.
Major comments
Figure 1. Induction of ISG54 by a series of 10-fold dilutions of SeV nucleic acids (RNA) extracted from the viral particles decreases progressively in a similar way with or without treatment of the viral preparations with DNase. Treatment with RNase abolishes ISG54 induction. This control expt is clear and unambiguous. Similarly, induction of ISG54 by the HBV nucleic acids from the viral particles (DNA +RNA) is totally inhibited by Dnase treatment, in line with its role on the DNA inducing pathway. However, treatment of the HBV nucleic acids (from viral particles) with RNase raises an interesting question. While RNase treatment does not affect the ability of the non-diluted preparation to induce ISG54, it is inhibitory at the 10-fold dilution.These data would suggest that the HBV DNA contained in the viral particles is organized in some structure that could resist RNase treatment when concentrated enough. A 10-fold dilution would desorganize this structure. Can the authors speculate on that ?
Figure 4. Drastic inhibition of ISG54 induction by transfecting the 10-fold dilution of HBV nucleic acids from the viral particles in the hepatocytes. Similar to my comment on the data in figure 1. Should have been compared to different dilutions of HT-DNA. If not done or possible to do, please comment.
Minor comments
Lines 284 and following: the authors refer to the ability of the DNA replicative intermediates to induce ISG54. However the color of the histograms showing these data on figure 1 is dark grey instead of black and is interpreted as RNase-treated samples. If this is a mistake, please correct. Along with this remark, it would be easier for the appreciation of the data to replace the black, dark and light grey colors throughout the figures by other colors (red, blue, yellow, etc).
Figure 6.The images from the two first rows are not well separated. Do the second and third rows of images correspond both to HBV+ HT-DNA? The third row gives an immediate impression of exclusion of the IRF3 from the nucleus when HBV is present, although the graphs presented on Figure 6B are clear and well explained. Maybe another way of presenting the data ? Adding other images of colocalization HBc and nuclear IRF3 upon HT-DNA transfection in HBV-infected cells ?
Line 133 titered instead of tittered
Round 2
Reviewer 1 Report
I appreciate the responsiveness to my comments and suggestion. The changes made in the revised manuscript are adequate and all statements are now supported by the results presented.